# Making the Most of Single Sensor Information: A Novel Fusion Approach for 3D Face Recognition Using Region Covariance Descriptors and Gaussian Mixture Models [note 1]

**DOI:** 10.3390/s22062388

**Published:** 2022-03-20

**Authors:** Janez Križaj, Simon Dobrišek, Vitomir Štruc

**Affiliations:** Faculty of Electrical Engineering, University of Ljubljana, Tržaška cesta 25, 1000 Ljubljana, Slovenia; simon.dobrisek@fe.uni-lj.si (S.D.); vitomir.struc@fe.uni-lj.si (V.Š.)

**Keywords:** face recognition, 3D images, local descriptors, statistical models

## Abstract

Most commercially successful face recognition systems combine information from multiple sensors (2D and 3D, visible light and infrared, etc.) to achieve reliable recognition in various environments. When only a single sensor is available, the robustness as well as efficacy of the recognition process suffer. In this paper, we focus on face recognition using images captured by a single 3D sensor and propose a method based on the use of region covariance matrixes and Gaussian mixture models (GMMs). All steps of the proposed framework are automated, and no metadata, such as pre-annotated eye, nose, or mouth positions is required, while only a very simple clustering-based face detection is performed. The framework computes a set of region covariance descriptors from local regions of different face image representations and then uses the unscented transform to derive low-dimensional feature vectors, which are finally modeled by GMMs. In the last step, a support vector machine classification scheme is used to make a decision about the identity of the input 3D facial image. The proposed framework has several desirable characteristics, such as an inherent mechanism for data fusion/integration (through the region covariance matrixes), the ability to explore facial images at different levels of locality, and the ability to integrate a domain-specific prior knowledge into the modeling procedure. Several normalization techniques are incorporated into the proposed framework to further improve performance. Extensive experiments are performed on three prominent databases (FRGC v2, CASIA, and UMB-DB) yielding competitive results.

## 1. Introduction

Face recognition systems are becoming increasingly popular due to their attractive properties such as high user acceptance, non-intrusiveness of the acquisition procedure and commercial potential in a diverse range of applications in both the private and public sectors [1,2]. The open issues in face recognition systems mainly relate to recognition in the presence of different sources of image variability, such as facial expressions and orientation, occlusions, illumination, time delay, or presence of makeup [3]. Such variability is ubiquitous in many applications, such as surveillance systems where images are captured under uncontrolled acquisition conditions and subjects are not cooperative [4]. To improve the reliability of the recognition procedure in the above scenarios, the use of 3D sensors to capture facial data has emerged as an important alternative to standard 2D cameras. The advantages of using 3D images for face recognition include invariance to lighting conditions and the ability to rotate 3D facial data into a normal pose [5], as well as providing additional information to defend against face spoofing attacks [6]. On the other hand, many 3D face recognition systems are still affected by facial expressions, occlusions and aging.

In addition to the use of standard 2D cameras, existing solutions for reliable face recognition include the use of multisensor approaches [7] as well as the use of specialized sensors such as 3D sensors [8], infrared assisted sensors such as FaceID [9] and Kinect [10], long-range sensors such as FaceSentinel [11], recent behind-the-screen sensors [12], thermal sensors [13], smart glasses [14] or multi-view sensors [15], to name a few. The main drawback of using special sensor hardware is its price, which may be prohibitive for many applications, while some laser scanner devices can also be harmful to human eyes, making them unsuitable for face recognition. Furthermore, some scanners have long acquisition times during which the face should remain still. When using multi-sensor approaches, it also proved difficult to obtain the optimal sensor combination based on the calibrated and fused information from the sensors, due to the heterogeneous sensors characteristics [7].

The main goal of using multiple or/and special sensors in the face recognition system is to provide additional information about the face to increase the robustness and the recognition performance of the system. Alternatively, the same goal can be pursued by acquiring face data with a single 2D or 3D sensor and then constructing a face recognition pipeline that ensures reliable performance. Existing approaches from this group include solutions applied at the data representation level [16], the data augmentation level [17], the feature extraction level [18], and the classification level [19]. Recently, deep neural network-based approaches for face recognition have become popular [20,21,22,23]. Approaches based on deep networks can combine all the above tasks from data representation to classification into a single end-to-end system. Such systems enable significant improvement in face recognition performance, but also require large amounts of training data.

The approach proposed in this paper uses a single 3D sensor in combination with the multiple (depth) data representations. Using a single depth sensor, we ensure fast acquisition times as well as the acquisition of detailed 3D shape information, while the different (3D) data representations add robustness to the face recognition process. The proposed face recognition system is fully automatic and proves to be robust to expression variations, partial occlusions, and moderate pose changes. Due to the robustness of our method, we only use a very simple and coarse face localization procedure. The alignment step is skipped and detection or removal of parts with occlusions/expressions is not required. We build on a framework for 3D face recognition previously proposed in [24] that capitalizes on region covariance matrixes (RCMs) and GMMs. The improvements over the previous framework include: (*i*) a novel face detection method that is more robust to occlusions since it does not rely on detection of any facial landmarks; (*ii*) inclusion of several data normalization techniques; (*iii*) thorough evaluation of recognition performance on the three challenging databases.

The work presented in this paper includes the following contributions: (*i*) A novel composite representation of 3D facial images based on various surface descriptors, such as shape index, Gaussian curvatures, surface normal coordinates, local binary patterns, etc. (*ii*) A novel local feature extraction method that unifies the above representations into a so-called composite representation, which is then used to extract local descriptors using covariance matrixes, transformation to Euclidean space, delta features, and PCA subspace projection. (*iii*) Integration of the above novelties into a new framework for fully automatic 3D face recognition that robust to image variability that occurs under real-world conditions, as shown by the experimental evaluation.

The paper is structured as follows. Section 2 summarizes related work. Section 3 contains a detailed description of the proposed framework. Section 4 presents the experimental evaluation and Section 5 concludes the paper with some final remarks.

## 2. Related Work

In this section, we outline a taxonomy of 3D face recognition techniques in terms of the type and number of sensors used to acquire the facial data. Specifically, in this section we capitalize on the difference between *single-sensors* techniques, which perform identity inference based on data coming from a single acquisition device, and *multi-sensor* solutions that combine information from several (typically diverse) acquisitions devices when performing identity recognition.

### 2.1. Single Sensor Techniques

3D face recognition methods can be categorized on the basis of the type of sensor they use to acquire facial data. According to the technologies used, depth sensors are normally categorized as active and passive devices. Essentially, all sensors in both categories acquire depth data using the triangulation principle. In active sensors, a triangle is defined between the light source, the object, and the sensor, while in passive sensors, the triangle can be formed between the object and two sensors [25]. Among the active sensors, the Minolta Vivid sensor is one of the most widely used for 3D face recognition. This sensor has been used to acquire multiple face image databases such as the Face Recognition Grand Challenge [26], Bosphorus [27], the CASIA [28], and the UMB-DB databases [29] to name a few. Methods for recognizing faces from images acquired by such sensors range from older subspace-based methods [30] to the latest state-of-the-art deep learning methods [20,31,32,33]. Instead of triangulation, low-cost active sensors typically use a structured light to compute depth, which provides much faster but less accurate and more noisy measurements [34]. Face recognition methods in [35,36] that use low-cost sensors such as Kinect pay special attention to removing noise from images. These methods often rely on representing the face through local features that are not affected by regional noise and distortions due to missing data, which are characteristic of low-cost sensors. The use of passive sensors for face recognition has the advantage of simplicity and applicability, since sensors of this type are built with relatively simple instrumentation [37]. Methods for recognizing faces from images acquired by passive sensors such as stereo cameras can obtain facial shapes from image sequences [38] or by fitting the estimated depth to a generic 3D model [39]. Recently, generative adversarial networks and deep convolutional networks have proven to be very successful in reconstructing facial shape and texture from a single 2D image [40].

Face recognition methods from the 3D sensors can also be grouped on the basis of the sensor data format. Depth sensors typically provide data in the form of a point cloud or in the form of a depth image. A point cloud is a set of data points, where each point contains information about its *x*, *y*, and *z* coordinates. Matching between point clouds is usually done by the iterative closest point algorithm (ICP) [41], which provides a dense point-to-point correspondence of 3D face shapes. If the points are projected onto the regular grid in the (x,y) plane, a depth image is obtained, which can be handled as a normal 2D grayscale image, where the value of each image pixel denotes the depth rather than the brightness. Consequently, many face recognition methods, such as subspace projection methods [42], originally developed for 2D images can be applied to depth images without much modification.

### 2.2. Multi-Sensor Modality

Some recognition methods use a combination of multiple sensors to acquire facial appearance data. The reasoning behind this is that multiple sensors provide more diverse data and that different sensors exhibit different characteristics under diverse environmental conditions. The most common multimodal approaches fuse information from 2D and 3D sensors, since many 3D sensors are also equipped with a calibrated 2D camera (e.g., Kinect, Minolta Vivid).

One of the first approaches of multimodal sensor face recognition in [43] investigates the comparison and combination of 2D and 3D face data for biometric recognition. It uses a PCA-based method tuned separately for 2D and 3D. They find no statistically significant difference between the recognition performance for both modalities, but report improved performance with a joined 2D – 3D solution when the fusion is performed at the classifier level.

In [7], the authors propose a face recognition system that integrates information from the visible, thermal-IR, and 3D time-of-flight sensors. When compared to the single sensor system, the proposed system shows improved performance on images with pose and illumination variations. They use the ICP algorithm to handle pose variations and various subspace projection methods for feature extraction.

Recently, an approach that uses several 2D sensors to capture images at multiple viewpoints was proposed in [44]. This approach incorporates the feature prior constraint and the texture constraint to explore the implied 3D information of uncalibrated multiview images of a person’s face.

## 3. The Proposed System

In this section, we describe basic characteristics of the 3D face recognition framework proposed in this paper, denoted as RCM_GMM_SVM for later convenience. We begin with a brief description of the entire framework and then explain in detail the preprocessing, data representation, feature extraction, modeling, classification, and normalization stages of the proposed approach. The section concludes by elaborating on characteristics of the proposed methodology.

### 3.1. Overview

Figure 1 shows a block diagram of the proposed 3D face recognition framework. The first procedural step of the framework involves the acquisition of a 3D face image. The data-acquisition step is followed by registration and preprocessing, which involves cropping the facial regionand filtering out all potential holes and spikes on the face images.

The next step is to map the preprocessed 3D facial data to a data structure, which we refer to as a *composite representation*. This composite representation is nothing more than different representations of 3D facial data stacked one upon another (see Figure 1). The composite representation is then analyzed in a block-by-block manner and a region covariance matrix (RCM) is extracted from all examined blocks. A local descriptor is obtained from each RCM matrix by transforming it in to Euclidean space using the Unscented Transform [45]. From these descriptors, delta coefficients are computed and their dimensionality is reduced by projecting them on to the PCA subspace. Finally, the descriptors are normalized to zero mean and unit variance.

Please note that unlike most other feature extraction techniques, RCM descriptors can be extracted from regions of variable sizes, allowing data to be examined from both local and holistic perspectives. Furthermore, RCM descriptors provide an elegant way to combine different representations of 3D data into a coherent feature vector.

After RCM extraction, each face is represented by several RCM descriptors whose distribution can be modeled by a GMM. Here, GMMs are selected for modeling purposes because they allow prior knowledge to be incorporated into the modeling procedure and naturally handle unreliable data. Each face can then be described by a so-called *supervector*, composed of the corresponding GMM parameters. Two normalization techniques are used at the supervector level, namely rank-normalization and within-class covariance normalization. Finally, an SVM-based classification scheme is used to classify the supervectors derived from the GMMs. At the end, normalization of the classifier scores is performed.

In the remainder of this section, we elaborate on all of the above steps and discuss their contribution to the robustness of the proposed recognition system.

### 3.2. Data Preprocessing and Localization

The input images are initially low-pass filtered to remove spikes. The *z* values (depth components) are interpolated and resampled uniformly on a grid with a resolution of 1.0 mm in the (x,y) plane. The face region is then localized on each preprocessed image.

The localization procedure (hereafter referred to as CB for Clustering Based) uses *k*-means clustering [46] to segment the 3D image into three (k=3) regions—background, body, and face (see Figure 2, where each color denotes one of the detected clusters). We choose the region with the lowest average depth as the face region. This procedure achieves only a rough localization of the facial region that may also include parts of a neck, hair and ears. The localized face is used as input for the subsequent recognition steps without any prior face alignment, occlusion removal, or normalization for facial expressions. However, these factors are addressed implicitly in the (local) feature extraction, the modeling, and the classification steps.

The CB localization method is computationally extremely simple and, due to the robust nature of the proposed system, more than sufficient to ensure satisfactory recognition results (see Section 4.3). In Figure 3, we see that the CB localization can reliably detect faces even under very challenging conditions where other, less robust localization methods often fail.

### 3.3. Data Representation

Let I represent a preprocessed depth image of size W×H. We then construct a W×H×D dimensional composite representation F from a given depth image I (see Figure 1) as follows
(1)F(x,y)=ϕ(I,x,y),
where the function ϕ extracts a *D*-dimensional vector f=F(x,y) from a pixel at position (x,y) of the image I. The vector f can be constructed by concatenating different representations of the image I at (x,y), including depth values, color information, pixel coordinates, image gradients, higher order derivatives, filter responses, differential-geometry descriptors, surface normals, etc.

In summary, a composite representation F represents a W×H×D tensor, where *W* and *H* represent the image width and height, and *D* denotes the number of representations combined in the tensor. A conceptual representation of the composite representation can be seen in Figure 1. It needs to be noted that there is no rule for how many or which 3D data representations should be combined into F for optimal face recognition performance. This issue has to be resolved experimentally and is addressed in Section 4.4.

### 3.4. Region Covariance Matrix

The composite representation F of a given 3D face image is analyzed locally block by block and from each block an RCM is constructed, from which feature vectors are eventually computed. Formally, any rectangular region R⊂F, comprising a set of vectors ${\left\{f_{n}\right\}}_{n=1}^{N}$, can be represented by a D×D covariance matrix [47]
(2)$$C_{R}=\frac{1}{N-1}\sum_{n=1}^{N}\left(f_{n}-\mu_{R}\right){\left(f_{n}-\mu_{R}\right)}^{T},$$
where $\mu_{R}$ is the mean vector of $f_{n}$. The diagonal entries of $C_{R}$ represent the variance of each feature and the non-diagonal entries represent their respective correlations.

Extracting the covariance of an inhomogeneous area results in a strictly symmetric and positive semidefinite matrix with constant dimensions that models the properties of the specified region. When no location-related representations (e.g., spatial coordinates) are used to construct the composite representation, the RCM descriptor is invariant to both rotation and scaling [47,48]. In this case, $C_{R}$ does not capture the ordering of the incorporated vector $f_{n}$ in the block/region R, nor the information regarding the size of the block from which it was extracted.

### 3.5. Unscented Transform

Covariance matrixes do not lie in Euclidean space (e.g., the covariance space is not closed under multiplication by negative scalars). Since most standard machine learning techniques are defined on Euclidean space, they are not directly applicable to work with covariance matrixes. Nonlinear mappings to Riemannian manifolds [49] or the Lie algebra [50] are therefore traditionally used to obtain vector spaces in which the metrics for machine learning methods are defined. This concept is also used in the Förstner metric [45], which approximates covariance dissimilarity measurement through log-manifold mapping and was originally proposed in [49] to measure the similarity between two RCMs. We cannot adopt the Förstner metric for our computations, since we plan to use the RCM-based feature vectors as input to the GMM-based modeling procedure and only then perform the matching procedure. Therefore, we consider a different approach based on the Unscented Transform (UT) [45,51].

The concept of UT is similar to Monte Carlo methods, with the difference that the vectors are not randomly generated. The UT encodes a given $C_{R}$ in a set of vectors ${\left\{w_{i}\right\}}_{i=1}^{2D+1}$ that, when treated as elements of a discrete probability distribution, have a covariance equal to a given $C_{R}$. The vectors $w_{i}$, unlike $C_{R}$, reside in Euclidean space and are defined as
(3)$$\begin{array}{cc} w_{0} & =\mu_{R},\\ w_{i} & =\mu_{R}+{\left(\sqrt{\alpha C_{R}}\right)}_{i},\;i=1\ldots D,\\ w_{i+D} & =\mu_{R}-{\left(\sqrt{\alpha C_{R}}\right)}_{i},\;i=1\ldots D, \end{array}$$
where ${\left(\sqrt{\alpha C_{R}}\right)}_{i}$ denotes the *i*-th column of the square root of the matrix $C_{R}$. The scalar α is a weighting factor for the elements in the covariance matrix and is set to α=2 in the case of the Gaussian distribution. To demonstrate the equivalence of the initial and the approximated distribution, we can compute an approximate sample mean vector $\mu_{R}^{\prime }$ and the corresponding covariance matrix $C_{R}^{\prime }$ by
(4)$$\mu_{R}^{\prime }=\frac{1}{2D+1}\sum_{i=0}^{2D}w_{i}\approx \mu_{R},$$
(5)$$C_{R}^{\prime }=\frac{1}{2D}\sum_{i=0}^{2D}\left(w_{i}-\mu_{R}^{\prime }\right){\left(w_{i}-\mu_{R}^{\prime }\right)}^{T}\approx C_{R}.$$

Each of the (2D+1) vectors $w_{i}$ resides in a *D*-dimensional Euclidean space, where $L^{2}$ distance computations can be performed. To obtain a single vector from each RCM, we concatenate all vectors $w_{i}$ extracted from a given RCM into one D(2D+1)-dimensional feature vector v:(6)$$v={\left[w_{0}^{T}w_{1}^{T}\ldots w_{2D+1}^{T}\right]}^{T}.$$

### 3.6. Delta Coefficients

Within the feature extraction procedure, we also include delta coefficients, which are commonly used in speech recognition. Deltas encompass the relations among the neighboring blocks and can therefore compensate for the assumption of feature vector independence in the subsequent GMM modeling step. Delta features thus integrate the interdependence among spatially adjacent vectors, since a different arrangement of vectors leads to different delta coefficients.

Given two *D*-dimensional feature vectors extracted from the neighboring blocks, i.e., $v_{i}=\left[v_{1}^{\left(i\right)},\ldots ,v_{D}^{\left(i\right)}\right]$ and $v_{i+1}=\left[v_{1}^{\left(i+1\right)},\ldots ,v_{D}^{\left(i+1\right)}\right]$, the *j*-th delta coefficient is defined as a difference between the features of neighboring blocks:(7)$$\Delta v_{j}=v_{j}^{\left(i+1\right)}-v_{j}^{\left(i\right)}.$$

Vertical delta features are computed from vertically adjacent blocks and horizontal delta features from horizontally adjacent blocks. The RCM-based feature vectors allow introduceingdepth deltas as well, the concept of which is presented in Figure 4. Depth deltas can be computed only due to the fact that the length of the RCM-based feature vectors does not depend on the size of the corresponding blocks.

All delta modalities provide us with feature vectors where the relations among adjacent blocks are implicitly included in the vectors themselves.

### 3.7. PCA Projection

Before statistical modeling, the feature vectors Δv are projected into the lower dimensional space using PCA. In this way, the redundant information in the feature vectors is eliminated and, at the same time, the computational complexity of the subsequent steps in our system is reduced. The PCA projection of a given feature vector v is defined as
(8)$$s=U^{T}\Delta v^{T},$$
where U denotes the eigenvector matrix computed offline using facial images from a training set. We determine the dimensionality of the projected feature vectors s experimentally, as described in Section 4.2.

### 3.8. Modeling

Next, the distribution of local feature vectors s is modeled by GMMs. Formally, a GMM can be defined as a superposition of *K* multivariate Gaussian probability density functions
(9)$$p\left(s\right)=\sum_{k=1}^{K}\pi_{k}N\left(s|\mu_{k},\Sigma_{k}\right),$$
where the parameters $\pi_{k}$ are called mixture weights and the Gaussian density $N(s|\mu_{k},\Sigma_{k})$ is a component of a mixture defined by its own mean $\mu_{k}$ and covariance $\Sigma_{k}$ as
(10)$$N\left(s|\mu_{k},\Sigma_{k}\right)=\frac{1}{{\left(2\pi \right)}^{N/2}{\left|\Sigma_{k}\right|}^{1/2}}\exp\left\{-\frac{1}{2}{\left(s-\mu_{k}\right)}^{T}\Sigma_{k}^{-1}\left(s-\mu_{k}\right)\right\}.$$
Given a set of descriptors $\Psi ={\left\{s_{n}\right\}}_{n=1}^{N}$, a GMM is constructed by determining its parameters based on maximizing the log-likelihood
(11)$$\log p\left(\Psi |\pi ,\mu ,\Sigma \right)=\sum_{n=1}^{N}\log\left\{\sum_{k=1}^{K}\pi_{k}N\left(s_{n}|\mu_{k},\Sigma_{k}\right)\right\}.$$

Maximum likelihood solutions (ML) for the model parameters are found via the Expectation-Maximization algorithm (EM) [52], initialized in our case by *K*-means clustering. When building user-specific GMMs (A user-specific GMM in this context is a GMM constructed from one 3D face image of a specific user.), there is usually not enough data available to reliably estimate the parameters of the GMM. Therefore, a universal background model (UBM) is typically constructed first, and then user-specific models are obtained by adaptating the UBM. A UBM is itself a GMM that represents generic, person-independent features. The parameters of the UBM are estimated via the ML paradigm (Equation 11) using all available training data. Once the UBM is built, user-specific GMMs are computed by maximum a posteriori (MAP) adaptation [53], adapting only the mean vectors ${\left\{\mu_{k}\right\}}_{k=1}^{K}$, by iteratively evaluating
(12)$${\hat{\mu }}_{k}=\left(1-\alpha \right)\mu_{k}+\alpha \mu_{k}^{EM},$$
where ${\hat{\mu }}_{k}$ is a new mean of the *k*-th Gaussian, $\mu_{k}$ is a mean from the previous step (initialized by the UBM), and $\mu_{k}^{EM}$ is the re-estimated mean from the M step of the EM algorithm. The parameter α balances the influence of the EM’s new statistics and the prior mean $\mu_{k}$ and is obtained for each component of the mixture as:(13)$$\alpha_{k}=\frac{N_{k}}{\tau +N_{k}},$$
where $N_{k}=\sum_{n=1}^{N}\gamma \left(z_{nk}\right)$ can be interpreted as the number of feature vectors assigned to the *k*-th mixture component and $z_{nk}$ is the posterior probability of *k*-th mixture component given a *n*-th feature vector.

Since the MAP adaptation preserves the order of the mixture components among different GMMs, all mean vectors ${\left\{\mu_{k}\right\}}_{k=1}^{K}$ from each user-specific GMM can be stacked in sequence to form the so-called supervector of a given 3D face image
(14)$$\rho ={\left[\mu_{1}^{T},\mu_{2}^{T},\ldots ,\mu_{K}^{T}\right]}^{T}.$$

Thus, each image is encoded by a single supervector of equal length.

### 3.9. Classification

Once the supervector is derived from the input 3D face image of a given user, it can be used to train an SVM classifier [54] for that specific user. SVMs are binary classifiers that seek a decision hyperplane with an our case, the supervectors ρ). It often happens that the classes are not linearly separable. Non-linear SVMs therefore project the input samples into a higher-dimensional space, where the samples can be linearly separated by a hyperplane. The SVM decision function takes the following form
(15)$$\nu \left(\rho \right)=\sum_{n=1}^{N}a_{n}t_{n}K\left(\rho ,\rho_{n}\right)+b,$$
where the coefficients $\alpha_{n}$ and *b* are the solutions of a quadratic programming problem [55] and $K\left(\rho ,\rho^{\prime }\right)=\phi {\left(\rho \right)}^{T}\phi \left(\rho^{\prime }\right)$ is a kernel function. It turns out that the kernel function can be computed without the explicit mapping ϕ(·) to the higher dimensional space, but requires only the computation of dot products between pairs of samples in the input (supervector) space.

During the enrollment stage, given a pool of supervectors ${\left\{\rho_{n}\right\}}_{n=1}^{N}$ from all *N* training images and the client’s supervector, the SVM training procedure constructs a decision hyperplane between the client’s supervector and the training supervectors. At test time, a supervector $\rho_{p}$ is first derived by MAP adaptation for the given client and the score $\nu (\rho_{p})$ is then computed. The value of ν denotes the distance of the client’s supervector to the decision hyperplane ν(ρ)=0 and can be treated as a dissimilarity measure that is eventually used for verification purposes.

### 3.10. Data Normalization

We apply several data normalization techniques on the feature vector, supervector, and matching scores levels to improve the recognition robustness of the framework. Data normalization in the image domain is already intrinsically included in the representations used to construct the composite representation. The normalization techniques used are described below.

**Zero-mean and unit variance normalization—MVN.** Within the descriptor domain, we standardize RCM-based descriptors to have zero mean and unit variance. Consider a descriptor s with its components $s_{i}$. For each component of the feature vector, we calculate the mean $\mu_{i}$ and standard deviation $\sigma_{i}$ using all feature vectors from the training set, and then normalize each component as
(16)$$s_{i}^{*}=\frac{s_{i}-\mu_{i}}{\sigma_{i}},$$
where $s_{i}^{*}$ stands for the normalized *i*-th component.

**Rank-based normalization—RN.** At the supervector level, a rank-based normalization is applied, where each component $\rho_{i}$ of the supervector ρ is replaced by the index (or rank) that the component would correspond to if the components of the *n* training images were arranged in ascending order
(17)$$\rho_{i}^{*}=\frac{{\mathrm{rank}}_{\rho_{1}\ldots \rho_{n}}(v_{i})-1}{n-1},$$
where $\rho_{i}^{*}$ is the *i*-th normalized component.

As a result of the rank-based normalization, the distribution of the supervector components in the normalized supervector $\rho^{*}$ approximates the uniform distribution.

**Within-class covariance normalization—WCCN.** In a ddition to RN, WCCN [56,57] is used at the supervector level. The WCCN, originally introduced in the context of SVM modeling [58], tries to minimize the expected classification error on the training data. To this end, the authors define a set of upper bounds on the classification error metric. By minimizing these bounds, the classification error is also minimized. The optimal solution to the minimization problem is given in terms of a generalized linear kernel, obtained by inverting the within-class covariance matrix $\Sigma_{W}$ computed as follows
(18)$$\Sigma_{W}=\sum_{i=1}^{N}\sum_{\rho_{j}\in \zeta_{i}}\left(\rho_{j}-{\hat{\mu }}_{i}\right){\left(\rho_{j}-{\hat{\mu }}_{i}\right)}^{T},$$
where $\rho_{j}$ denotes the *j*-th supervector of the *i*-th subject $\zeta_{i}$ in the training set and ${\hat{\mu }}_{i}$ is the mean of all supervectors of the *i*-th subject included in the training set. To obtain a WCCN-normalized supervector $\rho^{*}$, each supervector ρ is pre-multiplied by an upper triangular matrix U
(19)$$\rho^{*}=U\rho ,$$
where U is obtained by Cholesky decomposition of the matrix $\Sigma_{W}^{-1}$, i.e., $\Sigma_{W}^{-1}=U^{T}U$.

**Score normalization—SN.** Finally, normalization of the matching scores between the probe and gallery images is performed. Each gallery and probe image is compared to several pseudo-impostors (i.e., images from the training set). From the obtained scores, the mean $\mu_{g}$ and standard deviation $\sigma_{g}$ of the scores for each gallery image are computed. The same is applied to the probe images, estimating the mean $\mu_{p}$ and standard deviation $\sigma_{p}$ for each probe image. Then, each score $\nu_{gp}$ is normalized as follows
(20)$$\nu_{gp}^{*}=\frac{\nu_{gp}-\mu_{gp}}{\sigma_{gp}},$$
where $\mu_{gp}$ is defined as
(21)$$\mu_{gp}=\frac{\mu_{g}\sigma_{p}^{2}+\mu_{p}\sigma_{g}^{2}}{\sigma_{g}^{2}+\sigma_{p}^{2}}$$
and $\sigma_{gp}$ is defined as
(22)$$\sigma_{gp}=\sqrt{\frac{\sigma_{g}^{2}\sigma_{p}^{2}}{\sigma_{g}^{2}+\sigma_{p}^{2}}}.$$

The derivation of the (Equation 21) and (Equation 22) can be found in [59]. The effects of the above normalization techniques on verification performance are experimentally evaluated in Section 4.6.

### 3.11. Characteristics of the Proposed Approach

The proposed framework, summarized in Figure 1, has several desirable characteristics that ensure robust and effective recognition performance, as also demonstrated later in the experimental section, i.e.:*(i)* 
RCM descriptors are able to elegantly combine various face representations into a single coherent descriptor and can be considered as an efficient data fusion/integration scheme.*(ii)* 
RCM descriptors do not encode information about the arrangement or number of feature vectors in the region from which they are computed, and thus can be made scale and rotation invariant to some extent, but only if appropriate feature representations are selected for the construction of the composite representation F (see, e.g., [47,48]).*(iii)* 
Since RCM descriptors are computable regardless of the number of feature vectors used for their computation, they can handle missing data in the feature extraction step (i.e., even in the presence of holes in the face scans or in regions near the borders of the face scans, the RCM descriptor is still computable). Please note that this is not the case for other local features commonly used with GMMs, such as 2D DCT features, which require that all elements of a rectangular image-block are present.*(iv)* 
The size of the RCM-derived feature vectors does not depend on the size of the region from which they were extracted. Feature vectors of the same size can therefore be computed from image blocks of variable size. Thus, RCM-based feature vectors enable a *multi-scale analysis* By the term *multi-scale analysis*, we refer to the fact that the face can be examined at different levels of locality up to the holistic level.) of the 3D face scans.*(v)* 
GMM-based systems treat data (i.e., feature vectors) as independent and identically distributed (i.i.d.) observations and therefore represent 3D facial images as a series of orderless blocks. This characteristic is reflected in good robustness to imperfect face alignment, moderate pose changes (The term *moderate pose changes* refers to the pose variability typically encountered with cooperating subjects in a 3D acquisition setup. Examples of such variability are, for example, illustrated in Figure 3), and expression variations, as shown by several researchers, e.g., [60,61].*(vi)* 
The probabilistic nature of GMMs makes it easy to include domain-specific prior knowledge into the modeling procedure, e.g., by relying on the universal background model (UBM).*(vii)* 
Image reconstructions from GMMs confirm that the representations are invariant to partial occlusions and moderate rotations.

## 4. Experiments

The following subsections provide an evaluation of the performance and robustness of the RCM_GMM_SVM method and a comparison with other state-of-the-art methods. We also implement some of the popular local and holistic methods for 3D face recognition, summarized in Table 1 and include them in the comparison. In addition to the performance evaluation, we also assess the time complexity and study the proposed approach from the generative point of view.

The experiments evaluate two types of recognition systems, namely verification and identification systems. The results of the verification experiments are presented in the form of Receiver Operating Characteristic (ROC) curves or in the form of the verification rate (true acceptance rate) at a 0.1% False Acceptance Rate (FAR), whereas the results of the identification experiments are reported in the form of rank-1 identification rates or with Cumulative Match Characteristic (CMC) curves.

### 4.1. Used Databases

To perform a thorough experimental evaluation of the proposed recognition system, we use three data bases in our experiments, i.e., FRGC v2 [26], UMB-DB [68], and CASIA [28]. With the experiments on the FRGC v2 we evaluate the recognition performance in the case of a large number of subjects with near-frontal orientations and large expression variations. UMB-DB is used to observe the robustness of the proposed method to occlusions, while the robustness to pose variations is evaluated using the CASIA data, base.

The images in the FRGC v2 database have minor variations in pose and major variations in facial expressions. FRGC v2 contains 4007 3D facial images of 466 subjects, with up to 22 images per subject. The images were acquired with a Minolta Vivid 910 (this type of scanner is also used in UMB-DB and CASIA), which uses triangulation with a laser stripe projector to create a 3D image of the face. The images may contain shape artifacts, such as deformed areas due to movement of the subject during scanning, nose absence, holes, small protrusions, and impulse noise.

The UMB-DB consists of 1473 images (3D + color 2D) of 143 subjects. This data base was collected with special attention to facial occlusions that may occur in the real world. There are 590 images with partially occluded facial areas by different objects, such as hair, eyeglasses, hands, hats or scarves. The occlusions cover, on average, 42% of the face area, with a maximum of about 84%.

The CASIA data base consists of 4624 images of 123 subjects. There are 37 or 38 images per person containing (single) variations in pose, expression, and illumination, as well as combined variations of expressions under illumination and pose changes.

### 4.2. Experimental Parameter Setting

There are several parameters for feature extraction, model training, and classification that must be properly set for optimal operation of the proposed framework. We set the parameters based on a simple optimization procedure over a small number of parameter values and use the selected values for the later experiments.

The verification rates of the RCM_GMM_SVM system under different parameter settings are shown in Table 2. The parameter values in bold are used in the subsequent experiments. When analyzing different block sizes and step sizes between neighboring blocks, it can be observed that using smaller blocks leads to lower performance features computed from small blocks are less descriptive due to limited surface variability. On the other hand, larger block and step sizes result in a reduced number of observations per face, which also leads to lower performance. When investigating the effects of PCA dimensionality, we vary the length of the RCM-based feature vectors from 15 to 40. The best performance is obtained using the first 35 PCA components, but we use only the first 25 PCA components in the following experiments to reduce the computational cost. To test the effect of training data on recognition performance, we gradually increase the number of images used to train the UBM, from 10 up to the entire FRGC v1 database. As expected, more training data leads to better performance. The highest number of mixture components we studied is 1024, where the best performance is also achieved. However, in the following experiments, we use only 512 mixtures to ease the computational burden—512 components offer a good trade-off between performance and computational complexity.

### 4.3. Robustness to Imprecise Localization

To test the robustness of the proposed method to localization errors, we implemented three face localization procedures in addition to the CB method presented in Section 3.2:*Nose tip alignment (NT)*. The technique automatically detects the nose tip of the 3D faces and then crops the data using a sphere with radius r=100, similar to what is described in [69];*Metadata localization (MD)*. The technique uses the metadata provided by the FRGC protocol for face localization, i.e., manually annotated eye, nose tip and mouth coordinates;*ICP alignment (ICP)*. The technique localizes the face scans by first coarsely normalizing the position of the 3D faces using the available metadata, and then applying the iterative closest point algorithm for fine alignment with the mean face model.

The ICP and MD localization techniques use manually annotated characteristic points of the facial images, whereas the NT and CB techniques are fully automatic. Three baseline techniques are used in this analysis in addition to the proposed RCM_GMM_SVM approach, i.e., PCA_EUC, GSIFT_SVM and SIFT_SIFTmatch. Table 3 denotes the performance degradation that occurs when automatic face localization is used. The localization techniques are sorted from left to right in an increasing rate of localization imperfections. We see that SIFT_SIFTmatch and RCM_GMM_SVM, the representatives of local methods, are more robust to localization errors that the holistic methods, PCA_EUC and GSIFT_SVM, while the proposed RCM_GMM_SVM has the most stable performance among the compared methods.

### 4.4. Composite Representation Selection

Table 4 summarizes the experiments that analyze appropriate data representations for constructing the composite representation. Several representations are assessed (some of which can be seen in Figure 5), such as pixel coordinates (x,y), depth values I, shape index values $I_{s}$, Gaussian curvature values $I_{g}$, mean curvature values $I_{m}$, minimum curvature values $I_{min}$, maximum curvature values $I_{max}$, surface normal coordinates $I_{nx}$, $I_{ny}$ and $I_{nz}$, local binary patterns $I_{lbp}$ and angle values $I_{\varphi }$ between surface normals and the average facial normal. If we look at Table 4, the first thing we notice is that different combinations of image representations lead to significantly different verification rates. Among the evaluated combinations, the highest verification rate across all three data bases is achieved by the following W×H×4 dimensional composite representation
(23)$$F=[I_{s}\;I_{nx}\;I_{ny}\;I_{nz}].$$

Somehow unexpectedly, composite representations with more face representations do not always outperform composite representations with fewer face representations. This fact suggests that complementary information needs to be included in the composite representation to improve recognition performance.

### 4.5. Contribution of UT Transform and Delta Features

In this set of experiments, we evaluate the contributions of the UT transform and delta features, both of which are integral parts of the proposed feature extraction process. Table 5 summarizes the advantages of using the UT transform to derive the feature vectors (see Section 3.5). When UT is not applied, the feature vectors are formed directly from the elements of the RCM. By doing so, the feature vectors violate the postulates of the Euclidean space, which leads to a decreased verification performance of the system. Improved recognition performance can be obtained by extending the feature vectors with the mean vectors $\mu_{R}$ derived from the composite representations. However, the best performance is achieved by relying on the UT transform and constructing the feature vectors as shown in (Equation 6).

The contributions of delta features to recognition performance are summarized in Table 6. We see that both, horizontal and vertical deltas increase verification rate, while the highest verification rate is achieved when using combined horizontal and vertical deltas.

### 4.6. Evaluation of Normalization Techniques

Here, we assess the effects of the data normalization techniques described in Section 3.10 on recognition performance. The results of these experiments are shown in Table 7. The case where no normalization is used is denoted as *∅* (without normalization). We see that all normalization techniques contribute to the improvement of the verification rate, while RN normalization of supervectors brings the greatest improvement in the verification rate. The highest verification rate is obtained when all normalization techniques are included in the framework (last column in Table 7). Furthermore, we observe that the normalization techniques are beneficial for other techniques as seen by the results for the GSIFT_SVM approach.

### 4.7. Comparative Assessment on the FRGC v2 Database

Here we provide a comparative performance analysis of the proposed method with the latest state-of-the-art methods that also use the FRGC v2 data base in their experiments. To accurately compare the performance of the methods, we follow the FRGC v2 experimental protocol, which provides a set of standard verification experiments and defines three data sets—a training set, a gallery set, and a probe set. The training set is used to build global face models. In our experiments, images from the FRGC v1 data based are used as training images. The gallery set contains images with known identities (intended for enrollment), whereas the probe set contains images with unknown identities presented to the system for recognition. The FRGC v2 protocol provides several masks defining gallery and probe sets. We use the ROC I, ROC II, and ROC III masks to examine verification performance in the presence of a time lapse between the gallery and probe images. The ROC I experiment refers to images collected within a semester, while the ROC II experiment refers to images collected within the same year and the ROC III experiment refers to images collected between semesters. The *all vs. all* verification experiment uses all 4007 images as galleries and probes, resulting in more than 16 million comparisons (note that in this experiment gallery and probe sets are actually identical). Other partitions, i.e., *neutral vs. neutral*, *neutral vs. non-neutral*, and *neutral vs. all* are based on the facial expression labels and are provided by the FRGC protocol.

Table 8 shows the verification rates of the examined methods at 0.1% FAR. We report the results of the competing methods based on what is given in the corresponding papers, and also provide results for some other popular (holistic and local) methods implemented specifically for the comparative evaluation. We see that the performance of the RCM_GMM_SVM method is comparable to the other state-of-the-art methods, while noting that the RCM_GMM_SVM method uses only an extremely simple procedure to localize the faces and skips the alignment step. This is in contrast to the methods in [70,71,72], which outperform our method in some of the experiments.

The highest verification rate was obtained by [21], but this result comes at the expense of higher computational cost. A higher verification rate is also stated in [72], but a different experimental setup is used there. The authors randomly select up to six images from each subject to form the gallery set, and use the remaining images as probe set images. For the subjects with less than or equal to six images, they randomly select one image from each subject for the probe set and the remaining images for the gallery set. Then they calculate the matching score for each pair of gallery and probe images. They repeat the random division of the data set into the gallery and probe sets many times to be confident that every two images in the data set are matched. Table 9 shows the results of the above experimental setup used in [72]. Using this procedure, we achieved a verification rate of 99.9% at 0.1% FAR with only four images per subject.

We set up the identification experiments according to the protocols in the literature, considering four partition modes: (*i*) A-A. The earliest image of each subject is used as a gallery image and subsequent images of these subjects are used as probes; (*ii*) N-A. The earliest neutral image of each subject is used as a gallery image and subsequent images are used as probes; (*iii*) N-N. The earliest neutral image of each subject is used as a gallery image and subsequent neutral images are used as probes; (*iv*) N-$\bar{N}$. The earliest neutral image of each subject is used as a gallery image and subsequent non-neutral images are used as probes. These partitions enable closed-set identification, in which each probe image has a match among the gallery subjects. A comparison of the achieved identification performance is shown in Table 10. Our RCM_GMM_SVM method obtains rank-1 recognition rates consistently above the 98%, which is comparable to the highest identification results on the FRGC v2 data base. We can also conclude that expression variations have little effect on the identification performance of the RCM_GMM_SVM method.

### 4.8. Comparative Assessment on the UMB-DB Database

The UMB-DB database is employed to test the effectiveness of the proposed approach in the presence of occlusions. We use the CB method to localize faces as in all previous experiments. Thus, the facial images are recognized without prior detection or removal of occluded parts in the preprocessing step.

Both verification and identification experiments are performed on the images from the UMB-DB database. The results of the verification experiments can be seen in Table 11, where we followed the experimental protocol defined in [29]. Table 12 shows the results of the identification experiments, where we partitioned the images into gallery and probe sets similar to [83]. Note that the training set consists of the remaining images not included in the gallery or probe sets.

As the results in Table 11 and Table 12 show, the proposed method exhibits robust performance, even in the presence of severe occlusions that are present in the UMB-DB. We see that the difference in recognition performance between the RCM_GMM_SVM and SIFT_SIFTmatch systems is not as significant as for the FRGC v2 database, since the SIFTmatch local feature matching strategy naturally performs well when occlusions are present in the input facial images, as shown previously in [74]. On the other hand, the holistic approach in GSIFT_GMM fails completely in the presence of occlusions, since occluded areas are here directly included in the holistic feature vectors.

It should be noted that the systems described in [29,83] remove the occluded facial parts already in the preprocessing step. Therefore, the recognition performance of these systems depends heavily on the correct detection of the occluded parts. The proposed system, on the other hand, does not require detection of occluded parts. Since the subject-specific GMMs are adapted from the UBM, the α parameter ensures adapting only the components already seen in the training data and included in the UBM. Thus, the features corresponding to occluded areas do not have much impact on the subject-specific GMM estimation.

### 4.9. Comparative Assessment on the CASIA Database

The third set of comparative performance assessments is performed on the CASIA database, where we analyze robustness to pose variations. There is no experimental protocol for this database, so we use examples from the literature as a guide. As in the previous experiments, we divide the CASIA data into three subsets. The training set contains images from the last 23 of the 123 subjects, while the gallery and probe sets are constructed as described in Table 13 and Table 14, where we take into account the expression and occlusion labels provided in the CASIA metadata.

As with the previous two databases, we perform identification and verification experiments on the CASIA database. From the experimental results in Table 13 and Table 14, it appears that the RCM_GMM_SVM system shows relatively robust performance in the presence of occlusions compared to the competing techniques evaluated in this experiment.

### 4.10. Reconstruction of 3D Face Images from GMMs

The overall robustness of the proposed system can be attributed to the use of local features and statistical models. However, the most important role in ensuring robustness against occlusions is attributed to the latter, i.e., the statistical models. By relying on the UBM and MAP adaptation, an adequate statistical model of a person can be built even from a poor representation of the face at the feature level. To clearly demonstrate this characteristic, we assess the robustness of the proposed system from a generative point of view.

By randomly sampling from the GMMs, it is possible to generate synthetic data in the feature space and subsequently generate facial images. To generate a synthetic face image by random sampling, we choose a *k*-th Gaussian component (we pick it with probability given by its mixing coefficient ${\left\{\pi_{k}\right\}}_{k=1}^{K}$) and then generate a sample feature vector from the chosen component. For each generated feature vector, we find the closest match among the feature vectors from the training images and construct a face image from the surface patches belonging to the matched training feature vectors (for this purpose, we previously stored the feature/patch pairs of all training images).

Using this procedure, we generate synthetic images from Figure 6, where the top images of a pair represent 3D facial images from the UMB-DB that were automatically localized using the CB technique, and the bottom images show the corresponding synthetic faces generated by random sampling. We can see that expression and orientation variations are excluded from the synthetic facial images, while regions not seen in the original images due to occlusions are restored in the generated synthetic images. We argue that such variations are eliminated by estimating the GMM parameters from the UBM via MAP adaptation. Using the $\alpha_{k}$ parameter from (Equation 12), only the components that were *seen* in the training data are adapted. For the feature vectors extracted from occluded areas the $N_{k}$ from (Equation 13) will be small for all *K* components of the mixture model. Consequently, the $\alpha_{k}$ parameter will have a smaller value, while the adaptation (Equation 12) will rely more heavily on the UBM.

It can also be seen in Figure 6j,k that GMM models obtained from the facial images of the same person contain similar data. As expected, the random sampling from the UBM generates an average face (shown in Figure 7).

### 4.11. Time Complexity

In the final set of experiments, we evaluate the time needed by our framework to verify a single probe image and compare it to the processing times of the comepting techniques. All experiments were performed on a PC with an Intel Xeon CPU @ 2.67 GHz and 12 GB RAM.The methods were implemented using Matlab and therefore could be significantly sped up if implemented using a compiled language such as C/C++. The results of this part of our assessment are shown in Figure 8. The time complexity of the RCM_GMM_SVM method ranks in the middle compared to the other techniques. Compared to the techniques that use SIFTmatch classification, the RCM_GMM_SVM method has a significantly faster comparison/classification step. This makes the RCM_GMM_SVM method more suitable for the identification task where each probe subject needs to be matched with every gallery subject. The relatively short computation time of the proposed framework also results from the fact that only a simple clustering-based technique is used to locate the faces. This can also be observed in Figure 9, which shows the runtimes of all assessed localization techniques.

## 5. Conclusions

This paper addressed robust face recognition from data acquired in uncontrolled/real conditions by a single depth sensor. In such scenarios, we have to cope with different sources of image variability, such as changes in orientation, scale, facial expressions, and occlusions. The fully automatic recognition system proposed in this papersolves the problems of face detection, feature extraction, statistical modeling, and classification. Each of these problems was approached with the intention of increasing the overall robustness to variations that can occur in realistic situations. As demonstrated by experiments on three popular databases, the system is able to achieve high recognition performance even under very challenging conditions, and compares favorably with other state-of-the-art 3D face recognition systems from the literature.

## Figures and Tables

**Figure 1 sensors-22-02388-f001:**
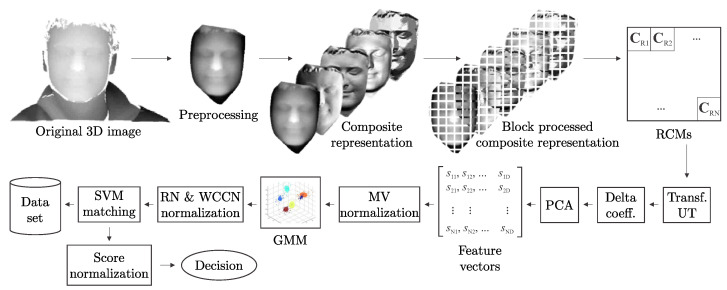
Conceptual diagram of the proposed system.

**Figure 2 sensors-22-02388-f002:**
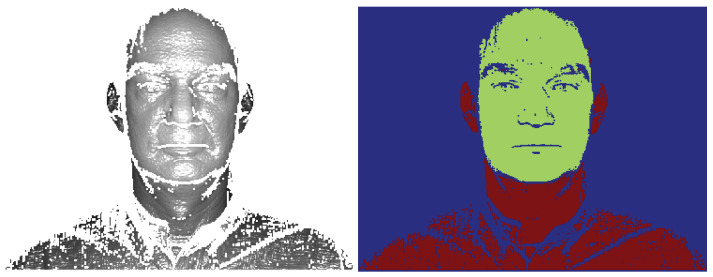
Input image (**left**) and the same image after CB localization (**right**).

**Figure 3 sensors-22-02388-f003:**
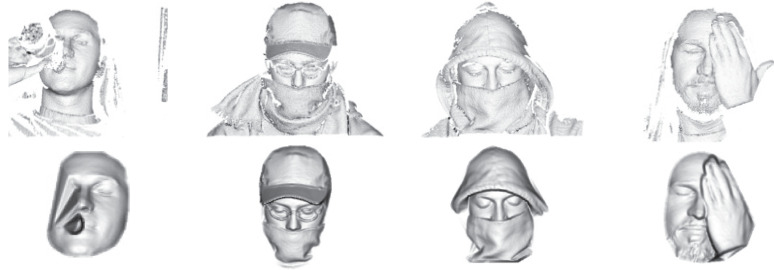
Original (**top**) and preprocessed (**bottom**) sample images from UMB-DB database.

**Figure 4 sensors-22-02388-f004:**
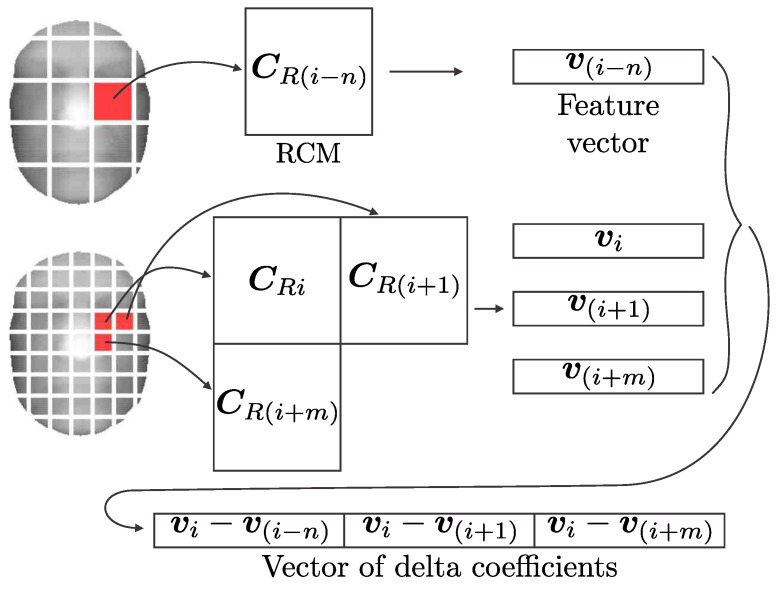
Schematic illustration of delta features extraction.

**Figure 5 sensors-22-02388-f005:**
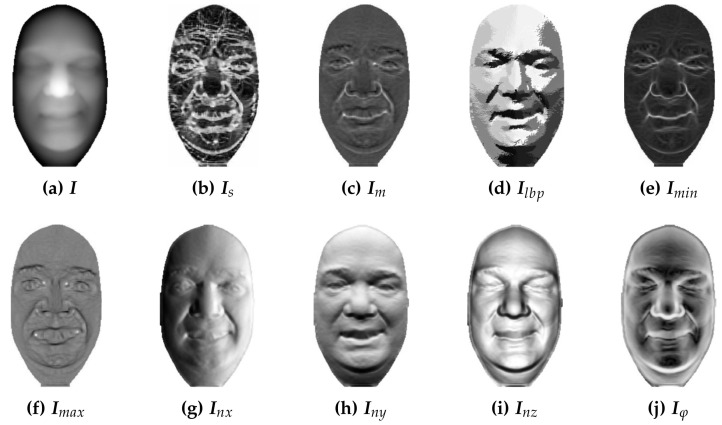
Different representations of depth data: (**a**) depth values, (**b**) shape index values, (**c**) mean curvature values, (**d**) local binary patterns, (**e**) minimum curvature values, (**f**) maximum curvature values, (**g**) *x*-values of surface normals, (**h**) *y*-values of surface normals, (**i**) *z*-values of surface normals, (**j**) angle values between surface normals and the average normal.

**Figure 6 sensors-22-02388-f006:**
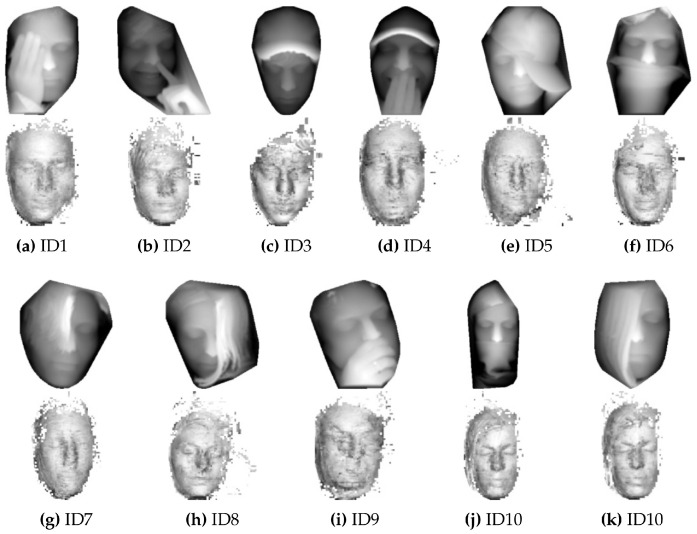
Preprocessed images (**top row**) and images generated by random sampling from the corresponding GMMs (**bottom row**).

**Figure 7 sensors-22-02388-f007:**
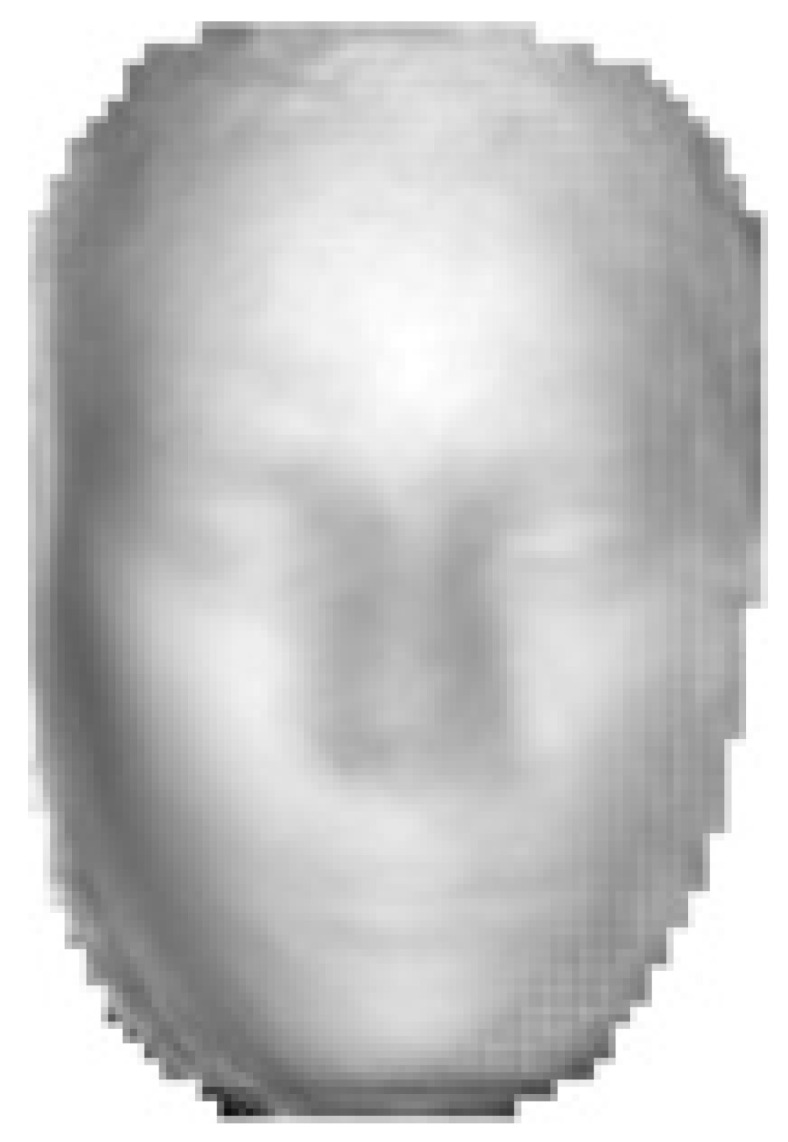
The image of an average face generated from the UBM.

**Figure 8 sensors-22-02388-f008:**
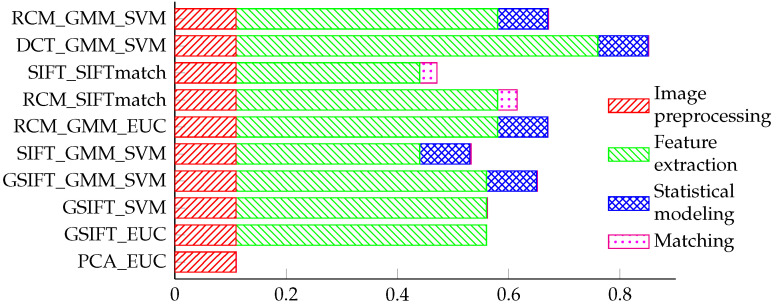
Average running times (in seconds) of the assessed methods for the verification of one probe image.

**Figure 9 sensors-22-02388-f009:**
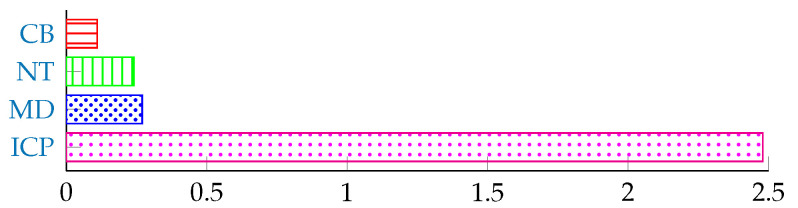
Average running times (in seconds) of the assessed localization techniques.

**Table 1 sensors-22-02388-t001:** Recognition methods implemented in this paper.

	Module		
Method	Feature Extraction	Feature Modeling	Classification
PCA_EUC [62]	PCA based holistic feature extraction	/	Euclidean distance-based similarity measure with nearest neighbor classifier
GSIFT_EUC [63]	SIFT ${}^{*}$ descriptors extracted from uniformly distributed locations on facial area	/	Euclidean distance-based similarity measure with nearest neighbor classifier
GSIFT_SVM [64]	SIFT ${}^{*}$ descriptors extracted from uniformly distributed locations on facial area	/	SVM
GSIFT_GMM_SVM [65]	SIFT ${}^{*}$ descriptors extracted from uniformly distributed locations on facial area	GMM	SVM
SIFT_GMM_SVM [65]	Classic SIFT ${}^{*}$ descriptors	GMM	SVM
SIFT_SIFTmatch [66]	Classic SIFT ${}^{*}$ descriptors	/	SIFT matching
DCT_GMM_SVM [67]	DCT-based descriptors	GMM	SVM
**RCM_GMM_SVM**	**RCM-based descriptors**	**GMM**	**SVM**

SIFT ${}^{*}$ descriptors extracted from the shape index representations of depth images.

**Table 2 sensors-22-02388-t002:** Verification rate (%) at 0.1% FAR under different parameter settings.

Parameter	Parameter Value/Verification Rate					
Block size (pixels)	15/95.6	20/95.7	**25**/96.1	30/95.1	35/92.1	40/91.9
Step size (pixels)	3/96.2	**4**/96.1	5/95.0	6/92.4	7/88.9	8/83.2
Feature vector length (no. of PCA comp.)	15/95.0	20/95.7	**25**/96.1	30/96.1	35/96.2	40/95.8
No. of training images to build the UMB	10/79.0	50/92.2	100/94.4	200/95.5	400/95.9	**943**/96.1
No. of Gaussian mixtures	32/86.9	64/92.7	128/94.1	256/95.7	**512**/96.1	1024/96.1

**Table 3 sensors-22-02388-t003:** Verification rate (%) at 0.1% FAR for different localization techniques (FRGC v2 *all vs. all* experiment).

	Localization Technique			
Method	ICP	MD	NT	CB
PCA_EUC	41.1	38.4	38.1	18.6
GSIFT_SVM	72.3	71.1	70.3	61.4
SIFT_SIFTmatch	90.2	90.0	89.9	89.1
**RCM_GMM_SVM**	**97.9**	**97.9**	**97.8**	**97.7**

**Table 4 sensors-22-02388-t004:** Verification rate (%) at 0.1% FAR for different composite representations F.

	Experiment		
F	FRGC v2 *all vs.* *all*	UMB-DB *neut., n.-occl.* *vs. occl.*	CASIA *neut., front. vs.* *n-neut., front.*
$\left[I_{nx}\;I_{ny}\;I_{nz}\right]$	94.8	81.2	94.2
$\left[X\;Y\;I_{nx}\;I_{ny}\;I_{nz}\right]$	95.8	83.5	93.6
$\left[I_{s}\;I_{nx}\;I_{ny}\;I_{nz}\right]$	**95.7**	**84.7**	**94.8**
$\left[X\;Y\;I_{s}\;I_{nx}\;I_{ny}\;I_{nz}\right]$	94.7	83.9	93.2
$\left[I_{lbp}\;I_{s}\;I_{nx}\;I_{ny}\;I_{nz}\right]$	93.6	82.0	92.1
$\left[I_{s}\;I_{g}\;I_{m}\;I_{min}\;I_{max}\right]$	92.3	82.3	81.4
$\left[X\;Y\;I_{s}\;I_{\varphi }\;I_{lbp}\right]$	78.3	65.6	77.2

**Table 5 sensors-22-02388-t005:** Influence of the unscented transform (UT) on the verification rate (FRGC v2 *all vs. all* experiment). The results represent the Verification rate (%) at a 0.1% FAR.

	UT Modality		
Method	Without UT	Without UT, Added $\mu_{R}$	With UT
RCM_GMM_SVM	93.9	94.7	96.1

**Table 6 sensors-22-02388-t006:** Influence of delta features on verification rate (FRGC v2 *all vs. all* experiment). The results represent the Verification rate (%) at a 0.1% FAR.

	Delta Features			
Method	Without	Horizontal	Vertical	Horizontal + Vertical
RCM_GMM_SVM	94.9	95.8	95.9	96.1

**Table 7 sensors-22-02388-t007:** Verification rate (%) at 0.1% FAR for different normalization techniques on the FRGC v2 *all vs. all* data set.

	Normalization Technique				
Method	∅	MVN	MVN + RN	MVN + RN + WCCN	MVN + RN + WCCN + SN
GSIFT_SVM	47.1	50.5	57.9	61.3	61.4
**RCM_GMM_SVM**	**81.0**	**84.0**	**96.1**	**97.5**	**97.7**

**Table 8 sensors-22-02388-t008:** Comparison with the state-of-the-art (verification rates (%) at 0.1% FAR on the FRGC v2).

	Experiment						
Method	*all vs*. *all*	*neut. vs*. *all*	*neut. vs*. *neut.*	*neut. vs*. *n.-neut*.	ROC I	ROC II	ROC III
Drira et al., 2013 [73]	94.0	n/a	n/a	n/a	n/a	n/a	97.1
Huang et al., 2012 [74]	94.2	98.4	99.6	97.2	95.1	95.1	95.0
Cai et al., 2012 [75]	97.4	n/a	98.7	96.2	n/a	n/a	n/a
Al-Osaimi et al., 2012 [76]	n/a	n/a	99.8	97.9	n/a	n/a	n/a
Queirolo et al., 2010 [77]	96.5	98.5	100.0	n/a	n/a	n/a	96.6
Kakadiaris et al., 2007 [78]	n/a	n/a	n/a	n/a	97.3	97.2	97.0
Wang et al., 2010 [70]	98.1	98.6	n/a	n/a	98.0	98.0	98.0
Inan et al., 2012 [71]	98.4	n/a	n/a	n/a	n/a	n/a	98.3
Mohammadzade et al., 2013 [72]	99.2	n/a	99.9	98.5	n/a	n/a	99.6
Emambakhsh et al., 2016 [79]	n/a	n/a	n/a	n/a	n/a	n/a	93.5
Soltanpour et al., 2016 [80]	99.0	99.3	99.9	98.4	n/a	n/a	98.7
Ratyal et al., 2019 [21]	99.8	n/a	n/a	n/a	n/a	n/a	n/a
Cai et al., 2019 [81]	n/a	100	100	100	n/a	n/a	100
Zhang et al., 2022 [82]	n/a	99.6	100	99.1	n/a	n/a	n/a
GSIFT_EUC	49.6	52.3	55.2	46.7	52.8	50.7	48.4
GSIFT_SVM	61.4	64.1	66.2	59.8	64.9	62.6	60.2
GSIFT_GMM_SVM	65.6	67.7	70.0	63.1	67.6	66.0	64.1
SIFT_GMM_SVM	77.3	83.7	94.4	69.9	78.1	77.0	75.9
RCM_GMM_EUC	82.7	91.2	97.9	83.7	84.3	83.1	81.8
RCM_SIFTmatch	87.5	91.9	97.1	82.4	88.0	87.6	87.2
SIFT_SIFTmatch	89.1	92.5	98.7	85.3	89.6	89.2	88.1
DCT_GMM_SVM	93.3	96.1	98.9	93.2	94.6	93.8	93.1
**RCM_GMM_SVM**	**97.7**	**99.2**	**99.8**	**98.5**	**98.6**	**98.1**	**97.7**

**Table 9 sensors-22-02388-t009:** Verification rate (%) at 0.1% FAR for different maximum numbers of images per gallery subject.

	Max. Number of Images per Gallery Subject					
Method	1	2	3	4	5	6
Mohamadzae et al. [72]	n/a	90.6	98.4	99.2	99.5	99.6
**RCM_GMM_SVM**	**97.7**	**99.6**	**99.8**	**99.9**	**99.9**	**99.9**

**Table 10 sensors-22-02388-t010:** Comparison with the state-of-the-art (Rank- 1 identification rate (%) on the FRGC v2).

	Experiment			
Method	A-A ${}^{*}$	N-A ${}^{†}$	N-N ${}^{‡}$	N-$\bar{N}$ ${}^{§}$
Drira et al., 2013 [73]	97.0	n/a	n/a	n/a
Huang et al., 2012 [74]	n/a	97.6	99.2	95.1
Cai et al., 2012 [75]	98.2	n/a	n/a	n/a
Al-Osaimi et al., 2012 [76]	97.4	n/a	99.2	95.7
Inan et al., 2012 [71]	97.5	n/a	n/a	n/a
Wang et al., 2010 [70]	98.2	98.4	n/a	n/a
Queirolo et al., 2010 [77]	98.4	n/a	n/a	n/a
Kakadiaris et al., 2007 [78]	97.0	n/a	n/a	n/a
Emambakhsh et al., 2016 [79]	n/a	97.9	98.5	98.5
Soltanpour et al., 2016 [80]	n/a	96.9	99.6	96.0
Ratyal et al., 2019 [21]	99.6	n/a	n/a	n/a
Cai et al., 2019 [81]	n/a	100	99.9	99.9
Yu et al., 2020 [31]	98.2	n/a	n/a	n/a
Zhang et al., 2022 [82]	99.5	n/a	n/a	n/a
SIFT_SIFTmatch	89.4	91.2	96.1	85.3
DCT_GMM_SVM	94.8	96.8	98.6	94.6
**RCM_GMM_SVM**	**98.1**	**98.9**	**99.6**	**98.2**

${}^{*}$ earliest as galleries, remaining as queries. ${}^{†}$ earliest neutral as galleries, remaining as queries. ${}^{‡}$ earliest
neutral as galleries, remaining neutral as queries. ${}^{§}$ earliest neutral as galleries, non-neutral as queries.

**Table 11 sensors-22-02388-t011:** Equal-error rates (%) on the UMB-DB (Values in the brackets are verification rates (%) at 0.1% FAR).

Subset			Method			
**Gallery**	**Probe**	**Training**	**Colombo** **et al**. [29]	**GSIFT_** **GMM_** **SVM**	**SIFT_** **SIFTmatch**	**RCM_** **GMM_SVM**
*neut., n.-occl.*	*neut., n.-occl.*	*n.-neut., n.-occl.*	1.9	4.8 (90.4)	0.8 (99.2)	**0.6 (99.2)**
*neut., n.-occl.*	*n.-neut., n.-occl.*	*occl.*	18.4	9.7 (63.6)	5.0 (90.2)	**3.0 (93.8)**
*neut., n.-occl.*	*neut., occl.*	*n.-neut., n.-occl.*	n/a	31.4 (11.5)	7.2 (79.1)	**3.6 (85.8)**
*neut., n.-occl.*	*occl.*	*n.-neut., n.-occl.*	23.8	34.9 (10.7)	7.9 (77.8)	**4.1 (84.7)**

**Table 12 sensors-22-02388-t012:** Rank-1 identification rate (%) on the UMB-DB.

	Experiment		
Method	N¯O-$\bar{O}$ ${}^{*}$	N¯O-¯N¯O ${}^{†}$	N¯O-O ${}^{‡}$
Alyuz et al., 2013 [83]	97.3	n/a	73.6
Ratyal et al., 2019 [21]	99.3	n/a	n/a
Xiao et al., 2020 [84]	n/a	n/a	61.6
GSIFT_GMM_SVM	92.3	76.0	21.2
SIFT_SIFTmatch	99.0	93.0	90.8
**RCM_GMM_SVM**	99.7	97.9	91.8

${}^{*}$ gallery: earliest neut. n.-occl.; probes: remaining n.-occl. ${}^{†}$ gallery: earliest neut. n.-occl.; probe: remaining n.-neut. n.-occl. ${}^{‡}$ gallery: earliest neut. n.-occl.; probe: occl. images.

**Table 13 sensors-22-02388-t013:** Verification rate (%) at 0.1% FAR on the CASIA database (Pose variations larger than 30${}^{\circ }$ are discarded).

Subset		Method	
**Gallery**	**Probe**	**SIFT_** **SIFTmatch**	**RCM_** **GMM_SVM**
*neut., front.*	*neut., front.*	98.8	**98.8**
*neut., front.*	*n.-neut., front.*	95.8	**96.9**
*neut., front.*	*neut., n.-front.*	59.5	**66.0**
*neut., front.*	*n.-neut., n.-front.*	53.0	**59.3**
*neut., front.*	*all*	77.6	**74.3**

**Table 14 sensors-22-02388-t014:** Rank-1 identification rate (%) on the CASIA database.

	Method				
Probe	Xu et al., 2009 [85]	Xu et al., 2019 [32]	Dutta et al., 2020 [86]	SIFT_ SIFTmatch	RCM_ GMM_SVM
IV(400) ${}^{*}$	98.3	n/a	98.2	99.3	**99.5**
EV(500) ${}^{†}$	74.4	n/a	n/a	97.6	**98.8**
EVI(500) ${}^{‡}$	75.5	99.1	n/a	98.2	**99.2**
PVS(700) ${}^{§}$	91.4	n/a	88.8	83.3	**85.3**
PVL(200) ${}^{¶}$	51.5	n/a	n/a	55.5	**59.5**
PVSS(700) ${}^{\|}$	82.4	n/a	n/a	76.7	**80.3**
PVSL(200) ${}^{\#}$	49.0	n/a	n/a	48.0	**51.5**

${}^{*}$ Illumination variations: top, bottom, left and right lighting. ${}^{†}$ Expression variations: smile, laugh, anger,
surprise and closed eyes. ${}^{‡}$ Expression variations under the lighting from the right side. ${}^{§}$ Small pose variations,
including views of front, left/right 20–30°, up/down 20–30° and tilt left/right 20–30°. ${}^{¶}$ Large pose variations,
including views of left/right 50–60°. ${}^{\|}$ Small pose variations with smiling. ${}^{\#}$ Large pose variations with smiling.

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
