# Peer review of "Making the Most of Single Sensor Information: A Novel Fusion Approach for 3D Face Recognition Using Region Covariance Descriptors and Gaussian Mixture Modelsâ€"

_sensors, 2022, doi:10.3390/s22062388_

Round 1
Reviewer 1 Report
In this work, the authors focus on face recognition from the images acquired by a single 3D sensor and propose a method based on the use of region covariance matrices and Gaussian mixture models (GMMs). Comprehensive experiments are performed on the three prominent data sets (FRGC v2, CASIA and UMB-DB) obtaining competitive results. However, the following problem need to be improved.
(1) The English of paper is poor, please polish it again.
(2) Each variable in the equation should be explained clearly.
(3) More recently references should be cited, such as
[*]Attention-Based Spatial-Temporal Multi-Scale Network for Face Anti-Spoofing, IEEE Transactions on Biometrics, Behavior, and Identity Science, vol. 3, no. 3, pp. 296-307, July 2021, doi: 10.1109/TBIOM.2021.3066983.
[**]Joint Face Alignment and 3D Face Reconstruction with Application to Face Recognition, IEEE Transactions on Pattern Analysis and Machine Intelligence, vol. 42, no. 3, pp. 664-678, 1 March 2020, doi: 10.1109/TPAMI.2018.2885995.
(4) More recently methods should be compared to show the advantages of the proposed method.
Author Response
We would like to thank the reviewer for the time and effort invested in reviewing our manuscript. We appreciate the constructive comments and feedback. We have done our best to incorporate all suggested changes and additions into the revised version of the manuscript. They are marked pink in the text. Detailed responses to the comments can be found below.
Comment: In this work, the authors focus on face recognition from the images acquired by a single 3D sensor and propose a method based on the use of region covariance matrices and Gaussian mixture models (GMMs). Comprehensive experiments are performed on the three prominent data sets (FRGC v2, CASIA and UMB-DB) obtaining competitive results. However, the following problem need to be improved.
Author's response: Thank you for the summary of our work.
Comment: The English of paper is poor, please polish it again.
Author's response: The revised version of the manuscript has been proofread by a native English speaker to improve the grammar and the readability. Parts of the manuscript have been rewritten to improve clarity. All grammar improvements are highlighted in blue in the revised manuscript to make it easier to identify the changes made.
Comment: Each variable in the equation should be explained clearly.
Author's response: We examined all equations included in the paper and made sure that all symbols are now explained in the text. This change resulted in smaller text revisions and additional lines of text that was added below Equations (16) and (17).
Comment: More recently references should be cited, such as
[*]Attention-Based Spatial-Temporal Multi-Scale Network for Face Anti-Spoofing, IEEE Transactions on Biometrics, Behavior, and Identity Science, vol. 3, no. 3, pp. 296-307, July 2021, doi: 10.1109/TBIOM.2021.3066983.
[**]Joint Face Alignment and 3D Face Reconstruction with Application to Face Recognition, IEEE Transactions on Pattern Analysis and Machine Intelligence, vol. 42, no. 3, pp. 664-678, 1 March 2020, doi: 10.1109/TPAMI.2018.2885995.
Author's response: We have incorporated the two references pointed out by the reviewer as well as some other recent ones in the revised manuscript.
Comment: More recent methods should be compared to show the advantages of the proposed method.
Author's response: Four recent methods were added to the performance comparison in Tables 7, 9, 11, and 13 and discussed accordingly in the text. The proposed approach is now compared against a considerable cross-section of recognition techniques across three diverse datasets.
Reviewer 2 Report
The manuscript, titled Making the Most of Single Sensor Information: A Novel Fusion Approach for 3D Face Recognition Using Region Covariance Descriptors and Gaussian Mixture Models, focused on face recognition from the depth image and propose a method based on the use of region covariance matrices and Gaussian mixture models. Some experimental results proved the proposed method is feasible and effective. It is a well written manuscript, a solid work with clear literature review, and the proposed method is described with many technique details.
I think it can be accepted to publish in Sensors after authors' minor revision.
- Authors claim Single sensor throughout, but lack the necessary support.
- The structure of this manuscript, especially the illustrations and tables, needs to be carefully considered and rearranged, which is difficult to read in current version, as well as the arrangement of sections, are the third-level section headings (e.g. 3.10.1~4) necessary?
- For Table 1, it is suggested that the original references of each method can be labeled for readers' understanding.
- How many blocks for Delta coefficients in the feature-extraction procedure is more suitable for optimal face recognition performance? It is better to make a further discussion;
- Some more experimental results about variable sized and moderate rotation image should be presented for well describe the characteristics of the proposed approach, and what is the exact meaning of moderate rotation?
- The headings of Sect. 4.2 is inappropriate. That is the experimental parameter setting, not the experimental setup of hardware device.
- Please provide the corresponding briefly discussion about the selection basis of No. of training images to build the UMB in Table 2, and a brief reason why not choose [x,y,Inx,Iny,Inz] for different composite representations in Table 4.
Author Response
We would like to thank the reviewer for the time and effort invested in reviewing our manuscript. We appreciate the constructive comments and feedback. We have done our best to incorporate all suggested changes and additions into the revised version of the manuscript. They are marked orange in the text. Detailed responses to the comments can be found below.
Comment: The manuscript, titled Making the Most of Single Sensor Information: A Novel Fusion Approach for 3D Face Recognition Using Region Covariance Descriptors and Gaussian Mixture Models, focused on face recognition from the depth image and propose a method based on the use of region covariance matrices and Gaussian mixture models. Some experimental results proved the proposed method is feasible and effective. It is a well written manuscript, a solid work with clear literature review, and the proposed method is described with many technique details.
I think it can be accepted to publish in Sensors after authors' minor revision.
Author's response: Thank you for the positive feedback on our work.
Comment: Authors claim Single sensor throughout, but lack the necessary support.
Author's response: The term single-sensor refers to the fact that the data used in the experiments comes from a single capture device. This is true for all databases used in the experiments, which all correspond to a single-sensor acquisition scenario. We agree that the term “single-sensor” may be ambiguous and could also be understood as “the same sensor”. We have therefore added a clarification to the introductory part of the Related work section, where we now explicitly define the meaning of the term “single sensor”.
Comment: The structure of this manuscript, especially the illustrations and tables, needs to be carefully considered and rearranged, which is difficult to read in current version, as well as the arrangement of sections, are the third-level section headings (e.g. 3.10.1~4) necessary?
Author's response: We have rearranged the tables and figures to be as close as possible to the paragraphs in which they are first mentioned. The placement specifier of figure and table environments has been changed from “t” to “H”, as suggested by the MDPI Latex template. We would like to note that the organization and placement of the figures and tables is subject to the copy editing from the publisher and will be adjusted in the final editing stage if the paper is accepted for publication.
In accordance with the suggestion made by the reviewer, we also removed the third-level section headers and replaced them with bold paragraph titles, as is standard in the computer vision literature these days.
Comment: For Table 1, it is suggested that the original references of each method can be labeled for readers' understanding.
Author's response: References to each of the methods in Table 1 were added as suggested.
Comment: How many blocks for Delta coefficients in the feature-extraction procedure is more suitable for optimal face recognition performance? It is better to make a further discussion.
Author's response: In Section 4.5, a discussion about the contribution of delta features has been added. A new table with additional experimental results on the impact of the delta features has also been included there.
Comment: Some more experimental results about variable sized and moderate rotation image should be presented for well describe the characteristics of the proposed approach, and what is the exact meaning of moderate rotation?
Author's response: The depth sensors used to collect the three datasets have a fixed depth range at which images were acquired. The depth images therefore exhibit a certain degree of variability in terms of size of the faces, defined by the operational range of the depth sensors. Similarly, as the goal of the datasets is to explore the performance of different recognition algorithms, most of the images are fairly frontal with a certain degree of variability as presented in the description of the datasets. We note that no labels or meta-data is included in the three datasets that would allow us to perform experiments with only specific pose variations or face sizes. Instead, the robustness across a certain type of variability is evaluated across the entire dataset. We agree that the term “moderate” need more clarifications, so we added an explanation to Section 3.11 to make this clearer.
Comment: The headings of Sect. 4.2 is inappropriate. That is the experimental parameter setting, not the experimental setup of hardware device.
Author's response: Thank you for pointing this out. We changed the title as suggested.
Comment: Please provide the corresponding briefly discussion about the selection basis of No. of training images to build the UMB in Table 2, and a brief reason why not choose [x,y,Inx,Iny,Inz] for different composite representations in Table 4.
Author's response: To comply with the FRGC experimental protocol, the UBM is trained on the FRGC v1 database, which contains 943 images. To test the effect of training data on recognition performance, we gradually increase the number of images used to train the UBM from 10 up to using the entire FRGC v1 database. As expected, more training data leads to better performance. We added a few sentences to the paper to explain this better.
We tested different possibilities for the composite representation and used the one that provided the best performance for all following experiments. The [x,y,Inx,Iny,Inz] would represent another potential choice with slightly lower performance.
Reviewer 3 Report
- I think, the English really required to be improved, such as the use of punctuation marks. For example, in the sentence "In this paper we focus on face recognition from...."
- The contributions and the novelties are not explicitly described in the introduction.
- The relationship between the equations in the paper is not explained in detail in an algorithm, so it is difficult for readers to understand it.
- The proposed method has not been tested with primary data in the real environment.
- The authors may be can cite the paper : S. -J. Horng, J. Supardi, W. Zhou, C. -T. Lin and B. Jiang, "Recognizing Very Small Face Images Using Convolution Neural Networks," in IEEE Transactions on Intelligent Transportation Systems, doi: 10.1109/TITS.2020.3032396.
Author Response
We would like to thank the reviewer for the time and effort invested in reviewing our manuscript. We appreciate the constructive comments and feedback. We have done our best to incorporate all suggested changes and additions into the revised version of the manuscript. They are marked red in the text. Detailed responses to the comments can be found below.
Comment: I think, the English really required to be improved, such as the use of punctuation marks. For example, in the sentence "In this paper we focus on face recognition from...."
Author's response: The manuscript has been carefully revised by a native English speaker to improve both grammar and readability. All grammatical improvements are highlighted in blue in the revised manuscript to make it easy to identify the changes.
Comment: The contributions and the novelties are not explicitly described in the introduction.
Author's response: A paragraph has been added to the introduction describing the main contributions of our work.
Comment: The relationship between the equations in the paper is not explained in detail in an algorithm, so it is difficult for readers to understand it.
Author's response: In addition to improving the grammar, we also tried to improve the integration of all equations in the manuscript and made sure that all variables introduced are properly discussed in the text. A similar comment was also made by another reviewer, so these changes are marked with different colors in the revised paper (pink, blue or red). Furthermore, in the summary section (3.11) we now point the reader back to the overview in Figure 1, with the aim of consolidating the descriptions of the overall algorithm.
Comment: The proposed method has not been tested with primary data in the real environment.
Author's response: We performed our experiments on publicly available databases that have been captured in real-world conditions and offer a common ground to compare performance with other competing approaches. Such an experimental evaluation provides a fair basis for comparison with other methods and follows standard methodology from established literature. The data in the datasets was captured in accordance with the operational characteristics of the utilized depth sensors (fixed range, cooperative subjects, etc.) and offers a representative sample of the expected variability that can be acquired with said sensors. Additionally, the datasets used have been captured with consenting subjects and IRB approval, which is critical for biometrics-oriented research. As with most papers on biometric recognition, the collection of novel datasets (ensuring GDPR compliance, IRB approval, etc.) is, therefore, beyond the scope of this paper.
Comment: The authors may be can cite the paper : S. -J. Horng, J. Supardi, W. Zhou, C. -T. Lin and B. Jiang, "Recognizing Very Small Face Images Using Convolution Neural Networks," in IEEE Transactions on Intelligent Transportation Systems, doi: 10.1109/TITS.2020.3032396.
Author's response: We have included the reference suggested by the reviewer in Introduction section of the revised manuscript.
Round 2
Reviewer 1 Report
The respond is OK.
Reviewer 3 Report
Face recognition is very important. Instead of using so many other sensors, this paper focuses on 3D sensor, GMM and SVM to do face recognition.
After checking all responses made by the authors, this paper can be accepted in its current form.